# ARP2/3 Regulates Fatty Acid Synthesis by Modulating Lipid Droplets’ Motility

**DOI:** 10.3390/ijms23158730

**Published:** 2022-08-05

**Authors:** Pengxiang Zhao, Hongbo Han, Xiang Wu, Jian Wu, Zhuqing Ren

**Affiliations:** 1Key Laboratory of Agriculture Animal Genetics, Breeding and Reproduction of the Ministry of Education, College of Animal Science and Technology, Huazhong Agricultural University, Wuhan 430070, China; 2Hubei Hongshan Laboratory, Wuhan 430070, China

**Keywords:** lipid droplets, motility, Hela, ARP2, ARP3, MYH9

## Abstract

The breakdown of lipid droplets (LDs) provides energy and contributes to the proliferation and migration of cancer cells. Recent studies have suggested that motility plays a key role in LD breakdown. However, the molecular mechanisms underlying LD motility were poorly characterized. In this study, we examined the function of microfilament-associated proteins 2 and 3 (ARP2 and ARP3) in regulating LDs’ motility in Hela cells. ARP2/3 mediated the LDs’ physical contact with F-actin and promoted the recruitment of Myosin Heavy Chain 9 (MYH9). MYH9 regulated the LD content by binding with LDs and ARP2/3. The number of LDs and TG content was increased after MYH9 interfered. The genes related to FA-related genes and neutral lipid synthesis-related genes were significantly increased (*p* < 0.05) when ARP2 and ARP3 were overexpressed. Bioinformatic analysis indicated that the high expression of ARP2/3 was associated with a poorer prognosis in cervical squamous cell carcinoma (CSCC). This study showed the effect of cytoskeletal filaments on LD metabolism in cancer cells.

## 1. Introduction

Lipid droplets (LDs) are cytoplasmic organelles for lipid storage which are surrounded by a phospholipid monolayer and coated with proteins involved in lipid metabolism [1,2]. In addition to storing lipids, LDs perform a variety of functions, such as serving as a platform for the turnover of some proteins [3], participating in cellular stress [4], and playing an important role in maintaining the homeostasis of the intracellular environment. Recently, the accumulated LDs have been increasingly considered as a striking feature of various cancers, raising interest in this field [5]. A high rate of de novo lipid synthesis is shown in many cancer cells. LDs are an essential part of the energy supplement to tumor cells. A series of lipid metabolizing enzymes are the major components of LD’s surface proteins, which tightly regulate the biosynthesis and degradation of Triacylglycerol (TAG), Cholesterol (CE), and membrane phospholipids. Excess fatty acid synthesizes to TAG via Acyl-CoA Synthetase (ACS), Glycerol-3-Phosphate Acyltransferase 4 (GPAT4), and Diacylglycerol O-Acyltransferase (DGAT 1 and 2) generates LDs from the Endoplasmic Reticulum [6,7,8]. On the contrary, LDs can be hydrolyzed by Adipose Triglyceride Lipase (ATGL) and Hormone-Sensitive Lipase (HSL) to generate free fatty acids (FFA) [9], the latter generating energy via mitochondria or peroxisomes [10]. LDs can also be delivered to autophagosomes for lipophagy. During these processes, the movement of LDs to specific sites in the cytoplasm to interact with other organelles is crucial. 

The movement of LDs involves three forms: free diffusion, local Brownian motion, and directional motion. Among them, the most crucial movement is the directional motion along the cytoskeleton. Some studies had reported that Actin and Actin-binding proteins were associated with LD’s motility [11]. The local aggregation of Actin in U2OS cells was essential to regulate the separation of clustered LDs. When COS-7 cells were treated with Nocodazole to disassemble microtubules, the contact of LDs with other organelles was significantly decreased [12]. The movement of LDs is regulated by three different myosin families: in yeast, LDs are transported from the parent to the daughter cell by V-type myosins. The clustering and separation of LDs in U2OS cells are regulated by non-muscle myosin II, while the movement is regulated by myosin I [13]. A large number of cytoskeletal proteins have been identified in the LD’s proteome, however, the proteins that mediate LDs binding to microfilaments have not been identified [14,15].

Previous studies showed that LDs could move directionally along microfilaments and the motor proteins attached to the cytoskeleton provide power [16,17]. The Actin Related Protein 2/3 (Arp2/3) complex is a fundamental driver of microfilament extension and cell motility [18]. Arp2/3 promotes peri-mitochondrial F-actin aggregation and increases DRP1- and MIEF1/2-dependent mitochondrial fission [19]. A Deficiency in the Arp2/3 complex causes the disruption of the cytoskeletal system. In our study, we investigated the molecular mechanism of ARP2/3 regulating LD’s motility in Hela cells. We found that the N-terminal of ARP3 regulated the binding of LDs to microfilaments. This process facilitates the recruitment of Myosin Heavy Chain 9 (MYH9) to LDs, thus providing sufficient power for LD movement along microfilaments. Knockdown of ARP2/3 resulted in diminished LD motility and increased number and size of LDs. In addition, TCGA analysis showed that the high expression of ARP2/3 was associated with a poorer prognosis of cervical squamous cell carcinoma. Our study illustrates the interactions between the cytoskeleton and LDs in cancer cells, providing potential insights and molecular targets for cancer therapy.

## 2. Results

### 2.1. ARP3 Localized on Microfilaments and LDs

It had been reported that the ARP2/3 complex played an important role in the microfilament network, so we examined the subcellular localization of ARP3 in the cell. The results showed that LDs, ARP3, and microfilaments have fluorescent co-localization (Figure 1A), which suggests the possibility that ARP3 is involved in their binding, but the precise mechanism of the interaction between LDs and microfilaments is unclear. We first investigated whether this process is mediated by proteins. We prepared artificial LDs (aLDs) [20] and observed their binding to microfilaments in vitro, which reduced the interference of other proteins in the cell. The aLDs and isolated intracellular LDs (cLDs) [21] were incubated, respectively, with purified actin for 30 min in vitro and their binding was observed by fluorescence microscopy (Figure 1B). The results showed that the binding ratio of cLDs to microfilaments was significantly higher than that of aLDs (Figure 1D). The isolated LDs were coated with a variety of proteins which indicates that this process is protein-dependent. Subsequently, we isolated LDs from ARP3 knockdown-cells and incubated LDs with actin. The binding ratio of LDs to actin was significantly reduced after knocking down ARP3 (Figure 1C,E). This suggested that ARP3 mediated the binding of LDs with microfilaments. Next, ARP3 was truncated at 1-112aa, 113-227aa, and 228-367aa from secondary structure domain analysis (Figure 1F). The N-terminal, C-terminal, and mid-terminal truncated fluorescent vectors of ARP3 were constructed and the interaction domains were detected by immunofluorescence. These results are consistent with the N-terminal region of ARP3(1-112aa) being the key domain mediating the interaction between LDs and microfilaments (Figure 1G,H). We attempted to locate a more precise structure and found that the region 1-56aa was highly overlapping with the localization of microfilaments, but 57-112aa did not show better LD binding ability.

### 2.2. ARP2 and ARP3 Regulated the Expression Level of FA and Lipid Synthesis-Related Genes

Lipid accumulation has represented a new hallmark of cancer. ARP3 mediated the interaction between LDs and microfilaments which suggested that we investigate the effects of ARP2 and ARP3 on the expression levels of lipid metabolic-related genes, including the FA-related genes (SREBF1, FASN, SCD1, and ACSL3) and neutral lipid synthesis-related genes (PLINs, FITMs, DGATs, PPARγ, and FSP27). We first investigated the effect of the suppression of ARP2 and ARP3 expression on these genes. The expression of ARP2 and ARP3 was decreased by approximately 70% through interference (*p* < 0.05, Figure 2A). Then, we detected the expression levels of FA and lipid synthesis-related genes. SREBF1, FASN, SCD1, and ACSL3 were significantly increased in the interference group cells compared with the control cells (*p* < 0.05, Figure 2B) which indicated the enhancement of FA synthesis. Similarly, the genes related to neutral lipid synthesis (DGAT1, DGAT2, PINN2, and PPARγ) were also increased when ARP2 and/or ARP3 were suppressed (*p* < 0.05, Figure 2C). Next, we evaluated the effects of these genes. When ARP2 and ARP3 were overexpressed (Figure 2D), the genes related to FA-related genes and neutral lipid synthesis-related genes were significantly decreased (*p* < 0.05, Figure 2E,F) in contrast to the results of the interference experiment. 

To understand the molecular mechanism by which ARP2/3 regulates lipid metabolism, we measured the ASK1-p38-JNK and AKT signaling pathways. The results showed that the p-ASK1, p-p38, and p-JNK1/2 levels were increased when ARP2 or 3 were knocked down (Figure 2G,I), and the p-AKT level also increased. As expected, the phosphorylation levels of all these proteins were significantly reduced when ARP2/3 was over-expressed (Figure 2H,J). These results indicated that ARP2 and ARP3 regulate the ASK1-p38-JNK and AKT signaling pathways to modulate lipid metabolism.

### 2.3. MYH9 Interacted with ARP3 for LD Movement

The direct movement of LDs depended on the cytoskeleton motor proteins to provide power [22]. MYH9 is a cytoskeleton-associated motor protein. We hypothesized that MYH9 is involved in the movement of LDs. We co-expressed ARP2/3 with MYH9 fluorescent plasmid in cells and examined the subcellular localization of both. As expected, the results showed that there was fluorescent co-localization of ARP2/3 with MYH9; they were looped around the LDs (Figure 3A). To further verify the interaction between ARP2/3 and MYH9, we co-expressed ARP3 and MYH9 intracellularly, and the results showed that ARP3 and MYH9 interacted with each other by immunoprecipitation (Figure 3B). Then, we investigated the structural domain of ARP3 binding to MYH9. We constructed three ARP3 fragment fluorescent plasmids separately, and, by observing the subcellular localization of ARP3 with MYH9, the results showed that 228-367aa of ARP3 is the MYH9 binding structural domain (Figure 3C). To further validate the binding domain, we expressed the truncated fragment of ARP3 with a Flag tag in the cells and mobilized MYH9 by immunoprecipitation; the results showed that the binding structural domain of ARP3 to MYH9 was 228-367aa (Figure 3D). To investigate whether the binding of ARP3 to MYH9 was associated with LDs, we co-expressed PLIN2-EGFP and MYH9-MCHERRY fluorescent plasmids in Hela cells and detected their subcellular localization (Figure 3E). PLIN2 was a well-recognized LD marker and the results showed that PLIN2 co-localized with MYH9 (Figure 3F). This indicated that MYH9 localized on the LD surface and provides the power for LD movement.

### 2.4. MYH9 Regulated LDs Content

Since ARP3 interacted with MYH9 and ARP3 specifically localized to the surface of LDs, we hypothesized that ARP3 could promote the recruitment of MYH9 on the surface of LDs. To verify this hypothesis, we knocked down ARP3 in cells, extracted intracellular LDs fractions separately, and used WB analysis to verify the expression level of MYH9 (Figure 4A). The results showed that the expression level of the LDs fraction of MYH9 was significantly reduced after knocking down ARP3. This suggested that ARP3 promoted the recruitment of MYH9 by LDs. MYH9 was associated with LD movement, which might affect intracellular LD content and interactions. We next knocked down MYH9 in cells and examined the level of intracellular LD changes (Figure 4B). The results showed that the knockdown of MYH9 increased intracellular LD content. The same results were obtained with OA treatment (Figure 4C,D), which further demonstrated that MYH9 could regulate LD content. Next, we measured intracellular triglyceride levels and the rate of triglyceride hydrolysis; the results showed that the knockdown of MYH9 cells had increased intracellular triglyceride levels and decreased triglyceride hydrolysis rates (Figure 4E,F), inspiring us to detect the interactions between LDs and mitochondria. As expected, LDs made fewer contacts with mitochondria (Figure 4G). These results suggest that MYH9 facilitates the movement of LDs along microfilaments.

### 2.5. The High Expression of ARP2 and ARP3 was Associated with the Poor Prognosis of Cervical Squamous Cell Carcinoma

Cervical Squamous Cell Carcinoma (CSCC) is a deadly gynecological cancer. We investigated the effect of the high expression of ARP2 and ARP3 on the prognosis of CSCC by analyzing the expression levels of ARP2 and ARP3 in CSCC via the GEPIA database. The data showed that ARP2 and ARP3 are highly expressed in CSCC (Figure 5A,B). In addition, the high expression of ARP2 and ARP3 in CSCC was associated with a poor prognosis (Figure 5C,D).

## 3. Discussion

The movement of LDs has a profound effect on the cellular state and cellular metabolism, and the consumption of LDs can provide energy to cells during a range of cellular life processes [23]. LDs exhibit various states of movement: free diffusion, restricted movement, and directed movement along the cytoskeleton. In early Drosophila embryos, almost all LDs move continuously back and forth along a straight line, showing no apparent free spreading [24]. In contrast, in the hepatocellular carcinoma cell line Huh7, LDs usually wobble within specific regions, occasionally switching to a state of rapid directional motion [25]. 

The directed movement of LDs is a universal phenomenon in various types of species or cells [26]. The movement increases the contact of LDs with other organelles, and the degradation or generation of LDs with other organelles facilitates various intracellular metabolic activities [27]. For example, in mitochondria, Milton and Miro1/2 mediate the recruitment of Dynin and Kinesin, and LDs move along microtubules as well as actin filaments [28]. The movement of LDs plays a crucial role in mitochondrial fusion, and LD movement promotes mitochondrial metabolism and enhances mitochondrial fitness [29]. LD movement is also closely related to lysosomes, and LDs play a crucial role in the micro-autophagy processes in which lysosomes are involved [30]. 

The movement of LDs requires the involvement of the intracellular skeleton and the dynamical system. However, the important proteins that mediate the binding of LDs to the cytoskeleton are not known [31]. A large number of cytoskeletal proteins are currently found in the LDs proteome [32]. When cells are starved, LDs can interact with mitochondria through directed movements along microtubules, and PLIN5 facilitates the contact between LDs and mitochondria. Mitochondria break down LDs for β-oxidation to provide energy to the cell [33]. This motivated us to focus on the role of cytoskeletal proteins in the movement of LDs.

ARP3 is a conserved microfilament protein of the actin-related protein family whose main function is to promote nucleation and assembly of microfilaments and facilitate cell migration [34,35]. The binding and movement of LDs to microfilaments increases lipid consumption, and when the binding is inhibited, LDs show reduced consumption and localized fusion of small LDs in the cell. In our study, we identified the 1-112aa region of Arp3 as the binding domain of LDs and microfilaments, where 1-56aa showed high agreement with microfilament localization. Interestingly, 57-112aa did not show a stronger LD binding ability and needs to be studied more precisely (Figure 1G,H). 

ARP2/3 not only affects the localization of LDs with microfilaments but also regulated the expression level of FA and lipid synthesis-related genes. The expression of SREBF1, FASN, ACSL3, DGATs, and PPARγ was significantly up-regulated after interfering with ARP2/3, indicating enhanced FA and neutral lipid synthesis (Figure 2). This is in agreement with a previous study. The SREBF1 and PPARγ signaling pathways are two classical pathways for intracellular lipid synthesis, and their activation can promote the activation of downstream lipid synthases and thus promote the synthesis of intracellular neutral lipids [31,36]. ARP2 and ARP3 regulate the expression of genes related to lipid metabolism. The binding and movement of LDs to microfilaments increases lipid consumption and, when the binding is inhibited, LDs show reduced consumption and localized fusion of small LDs within the cell [37]. MYH9 contains an IQ domain and a myosin head-like domain which is involved in several important functions, including cytokinesis, cell motility, and maintenance of cell shape. In our study, we found that MYH9 interacted with ARP2/3 and LDs, indicating that it might participate in the directional movement of LDs. After interfering with MYH9, cellular TG content was significantly increased, suggesting that MYH9 regulated cellular metabolism. We speculate that this was due to a decrease in the movement of LDs, but how MYH9 affects cellular TG content still requires further investigation. 

## 4. Materials and Methods

### 4.1. Cell Culture

The Hela cell line was gifted by the lab of Prof. Xianghua Yan, Huazhong Agricultural University (Wuhan, China) and was purchased from the Type Culture Collection of the Chinese Academy of Sciences (Wuhan, China). Hela cells were cultured in Dulbecco’s Modified Eagle Medium (DMEM; HyClone, Logan, UT, USA) with 10% fetal bovine serum (FBS; #SH30396.03, Hyclone, Montreal, QC, Canada), 100 unit/mL penicillin, and 100 µg/mL streptomycin in dishes at 37 °C, in a humidified atmosphere, with 5% CO_2_. For the oleic acid treatment, a 20 mM oleic acid-phosphate buffer saline (PBS) mixture and 20% FA-free bovine serum albumin (BSA) medium were prepared and both media heated in a 70 °C water bath for 30 min. Finally, the media were mixed. The 10 mM oleic acid-BSA mixture was added to the cell culture medium at 1:49 (v:v). The cells were then either seeded on slides or on plates that had been washed three times using PBS. Then, 1 mL oleic acid medium was added to the well, and the cells were cultured for 12 h. OA treatment was only used in Figure 4B,D.

### 4.2. Transfection Assay

Cells were seeded on a 6-well plate or slides in a 24-well plate. Then, the cells were transfected with Lipo8000™ Transfection Reagent (#C0533, Beyotime, Nanjing, China). For the preparation of RNAi working solution, 10 µL siRNA oligo (20 µM, Ribobio, Guangzhou, China) was mixed with 10 µL Lipo8000 regent in 100 µL DMEM. For the preparation of the overexpression working solution, 2.5 µg plasmid was mixed with 4 µL Lipo8000 regent in 50 µL DMEM. The working solution was added to the plate well and incubated for 6 h. Then, the plate well was changed with a fresh cultural medium (DMEM with 10% FBS) for another 48 h of culture.

### 4.3. Plasmid DNA Construction

For the overexpression assay and the localization assay, the expression vector and fluorescence-labeled vector were constructed. In brief, the ARP2/ARP3 CDS region was amplified by the cDNA library of Hela cells using KOD-Plus-Neo DNA polymerase (#KOD-401, TOYOBO, Shanghai, China). After gel extraction, the ARP2/ARP3 CDS fragment was cloned into the digested pcDNA3.1 vector (digestion sites, HindIII and BamHI) using a seamless cloning kit (#C112-01, ClonExpress II One Step Cloning Kit, Vazyme, Nanjing, China). For the localization assay, the gene CDS region was cloned into the digested pCMV-C-EGFP (#D2626, Beyotime Biotechnology, Nanjing, China).

### 4.4. Fluorescence Marking and Analysis

The cell slides were fixed with 4% paraformaldehyde for 15 min at room temperature. The slides were stained with different dyes for 10 min at 37 °C. LDs were stained with BODIPY 493/503 (#D3922, Invitrogen, Carlsbad, CA, USA) for green or LipidTox (#H34477, Invitrogen, Carlsbad, CA, USA) for deep red. DAPI (#G-1012, Servicebio, Wuhan, China) was used for the nucleus staining. Microfilaments were stained with Alexa Fluor™ 568 Phalloidin (#A12380, Invitrogen, Carlsbad, CA, USA). Mitochondria were stained with MitoTracker™ Red CMXRos (#M7512, Invitrogen, Carlsbad, CA, USA). After washing three times with PBS for 10 min each, the slides were sealed with an anti-fluorescent quenching solution (#P36961, ProLong™ Diamond Antifade Mountant, Invitrogen, Thermo Fisher, Waltham, MA, USA) for confocal microscopic observation (63× oil lens, BODIPY FL and DAPI channels, Zeiss LSM 800, Oberkochen, Germany).

Image analysis was performed in ImageJ(ImageJ 1.52a, NIH, Bethesda, MD, USA). The measure of interaction between mitochondria and LDs was performed by methods previously reported [38]. Briefly, mitochondria within 0.5 µm of the LD’s edge were defined as interaction while mitochondria beyond the 0.5 µm peridroplet region were defined as no-interaction.

### 4.5. Western Blot

Standard methods were used to perform Western blot. Briefly, cells were collected and homogenized in a lysis buffer (#P0013, Beyotime Biotechnology, Nanjing, China). Then, the homogenates were incubated with an SDS-PAGE sample loading buffer (#P0015A, Beyotime Biotechnology, Nanjing, China) at 98 °C for 10 min. Subsequently, the samples were separated by 10% sodium dodecyl sulfate-polyacrylamide gel electrophoresis (SDS-PAGE) and transferred to a polyvinylidene fluoride (PVDF) membrane (Biorad, Hercules, CA, USA) using a semidry electrophoretic apparatus. The blocked membranes (#P0252-100 mL, QuickBlock™ Blocking Buffer for Western Blot, Beyotime Biotechnology, Nanjing, China) were incubated with antibodies overnight at 4 °C. The blots were washed thoroughly three times with tris-buffered saline with tween20 (TBST) buffer for 10 min and incubated under gentle agitation with the primary antibodies for immunodetection at 37 °C for 1.5 h (diluted in QuickBlock™ Primary Antibody Dilution Buffer for Western Blot, #P0256, Beyotime Biotechnology, Nanjing, China). Then, the blots were extensively washed three times with TBST. Subsequently, blots were incubated under gentle agitation with the secondary antibodies for immunodetection at 37 °C for 1 h (diluted in QuickBlock™ Secondary Antibody Dilution Buffer for Western Blot, #P0258, Beyotime Biotechnology, Nanjing, China). For detection, a M5 eECL Western Blot Kit (#MF-078-01, Mei5bio, Beijing, China) and the chemiluminescence imaging system (LAS4000, ImageQuant, München, Germany) were used.

The rabbit polyclonal antibodies that were used included anti-Arp3 (A4514, Abclonal, Wuhan, China), anti-ASK1 (A3271, Abclonal, Wuhan, China), anti-p-ASK1 (AP0058, Abclonal, Wuhan, China), anti-JNK1 (A0288, Abclonal, Wuhan, China), anti-P-JNK1 (AP0631, Abclonal, Wuhan, China), anti-p38 (A4771, Abclonal, Wuhan, China), anti-p-p38 (AP0057, Abclonal, Wuhan, China), anti-AKT1 (A20799, Abclonal, Wuhan, China), anti-pAKT1 (AP1259, Abclonal, Wuhan, China), anti-Myh9 (A0173, Abclonal, Wuhan, China), and anti-GAPDH (A19056, Abclonal, Wuhan, China).

### 4.6. Fluorescence Image Analysis

ImageJ software(ImageJ 1.52a, NIH, Bethesda, MD, USA) was utilized to analyze the number of actin interacting LDs. Briefly, the fluorescence intensity was analyzed along with the dotted line (Figure 3C,D). If the LD contacts actin, the signal of CY3 (actin) can be detected between or overlapping the signal of BODIPY493/503 or Lipidtox (LDs). If the LD does not contact actin, the signal of CY3 (actin) is supposed to be lost between the signal of BODIPY493/503 (LDs).

### 4.7. In Vitro Binding Test

The artificial LDs were prepared according to the method reported previously. Briefly, 2 mg dried 1,2-di(9Z-octadecenoyl)-sn-glycero-3-phosphatidylcholine (DOPC) and 5 mg triglyceride were vortexed with 100 μL of buffer B and then centrifuged to collect the upper LDs. aLDs were incubated with the cytoskeleton or cell extracts together or alone at room temperature for 30 min, collected by centrifugation at 12,000× *g* for 5 min, and then rinsed twice with buffer B for the next experiment.

### 4.8. TG and TG Hydrolysis Measure

TG was measured by (#A110-1-1, Jiancheng, Nanjing, China) following the manufacturer’s recommendations. Briefly, collected cells were added to 200 μL PBS and sonicated under an ice water bath. The working solution and samples were added sequentially to a 96-well plate and mixed, then incubated for 10 min at 37 °C. The optical density at 510 nm was measured on a microplate reader (PerkinElmer, Rodgau, Germany). The TG hydrolysis rate was estimated by the measurement of glycerol release into the medium from isolated mature adipocytes over 90 min. Glycerol released in the medium was measured using a free glycerol determination kit (#FG0100, Merck, Darmstadt, Germany) following the manufacturer’s recommendations.

### 4.9. Isolation of Lipid Droplets

LDs were purified from Hela cells by methods previously reported [7]. Briefly, cells were broken up using a homogenizer with 500 μL Buffer A (25 mm tricine, 25 mm sucrose, pH 7.8) containing 1 mm PMSF and incubated on ice for 30 min. This was then centrifuged at 16,000× *g* for 1 h at 4 °C, and the upper white LDs were collected and washed with 500 μL of buffer B twice.

### 4.10. Survival Analysis

Survival prediction was performed using the GEPIA database (http://gepia.cancer-pku.cn/, accessed on 20 January 2022) according to the instructions of the creator of this database.

### 4.11. Statistical Analysis

All experiments were repeated at least three times. All results are presented as the mean ± SD, *n = 3*. Statistical significance was assessed using the Student’s *t*-test. A *p*-value < 0.05 was deemed to indicate statistical significance.

## 5. Conclusions

In our study, we observed that the N-terminal of ARP3 regulated the binding of LDs to microfilaments. ARP2/3 facilitated the recruitment of MYH9 to LDs, thus providing sufficient power for LD movement along microfilaments. The knockdown of ARP2/3 increased the number of LDs and TG content, but the cell proliferation rate was decreased. In addition, TCGA analysis showed that the high expression of ARP2/3 was associated with a poorer prognosis of cervical squamous cell carcinoma. Our study illustrates the interactions between the cytoskeleton and LDs in cancer cells, providing potential insights and molecular targets for cancer therapy.

## Figures and Tables

**Figure 1 ijms-23-08730-f001:**
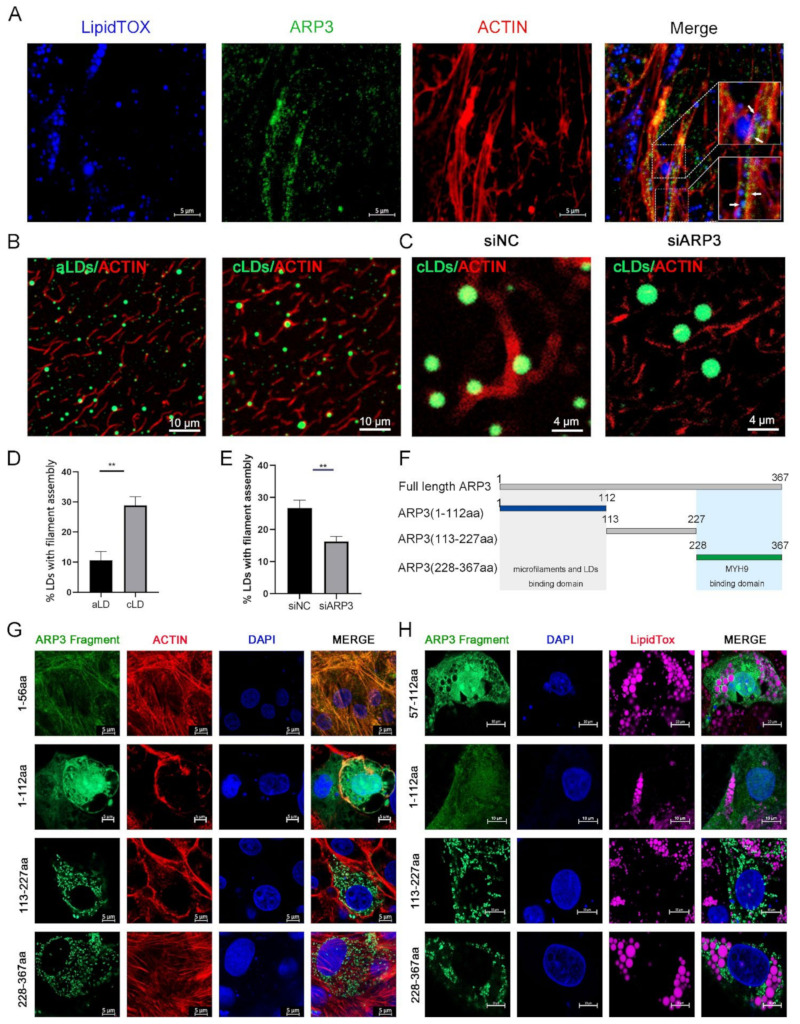
ARP3 localized on microfilaments and Lipid droplets (LDs). (**A**) Subcellular localization of ARP3 and LDs, bar, 10 μm. Arrow:co-location between LDs, Arp3, and microfilaments. (**B**) The binding of artificial LDs (aLDs) or intracellular LDs (cLDs) and actin, bar, 10 μm. (**C**) The binding of isolated intracellular LDs (cLDs) and actin after being treated with siNC or siARP3, bar, 4 μm. (**D**) The Quantitative Analysis of LDs binding with microfilaments of (**B**), ** *p* < 0.01. (**E**) The Quantitative Analysis of LDs binding with microfilaments of (**C**), ** *p* < 0.01. (**F**) Schematic diagram of ARP3 truncation fragments. (**G**) Subcellular localization of ARP3 truncation fragments and microfilament, bar, 5 μm. (**H**) Subcellular localization of ARP3 truncation fragments and LDs, bar, 5 μm.

**Figure 2 ijms-23-08730-f002:**
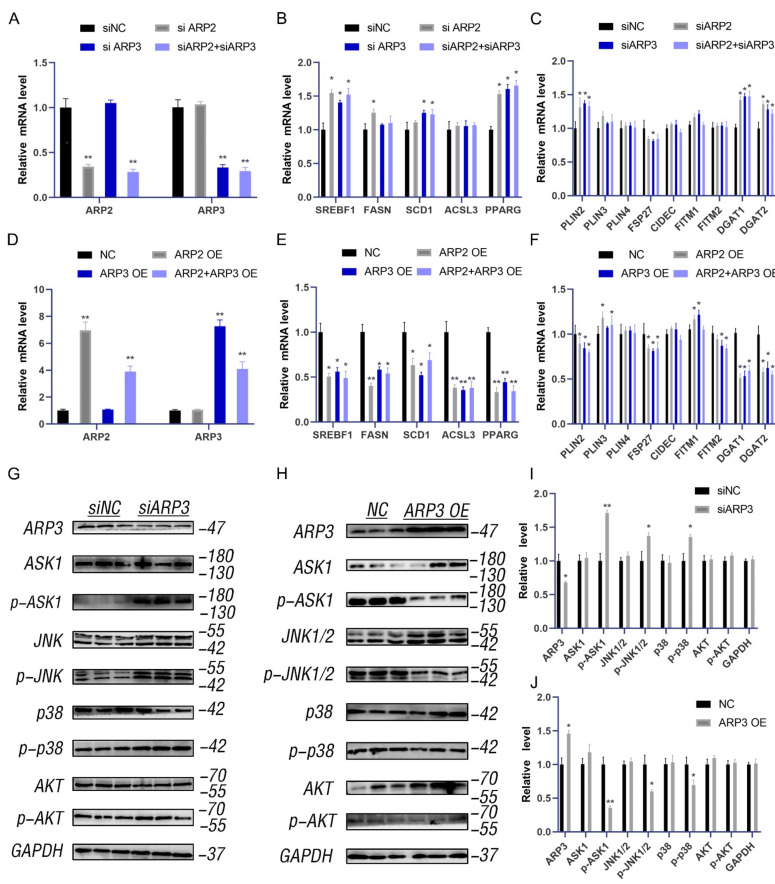
ARP2 and ARP3 regulate the expression level of FA and lipid synthesis-related genes. (**A**) Interference efficiency detection by qRT-PCR. * *p* < 0.05, ** *p* < 0.01 (**B**) The expression level of fatty acid synthesis-related genes detected by qRT-PCR. * *p* < 0.05. (**C**) The expression level of lipid synthesis-related genes detected by qRT-PCR. * *p* < 0.05. (**D**) Overexpression efficiency detection by qRT-PCR. ** *p* < 0.01. (**E**) The expression level of fatty acid synthesis-related genes detected by qRT-PCR. * *p* < 0.05. (**F**) The expression level of lipid synthesis-related genes detected by qRT-PCR. * *p* < 0.05. (**G**) The ASK1-p38-JNK and AKT signaling in groups of interfered ARP3 and control cells as detected by Western Blot experiments. (**H**) The ASK1-p38-JNK and AKT signaling in the overexpression cell group and control cells as detected by Western Blot experiments. (**I**) A grey value analysis of G. ImageJ software (ImageJ 1.52a, NIH, Bethesda, MD, USA)was used for this analysis, according to the instructions. * *p* < 0.05, ** *p* < 0.01. (**J**) A grey value analysis of H. ImageJ software (ImageJ 1.52a, NIH, Bethesda, MD, USA)was used for this analysis, according to the instructions. * *p* < 0.05, ** *p* < 0.01.

**Figure 3 ijms-23-08730-f003:**
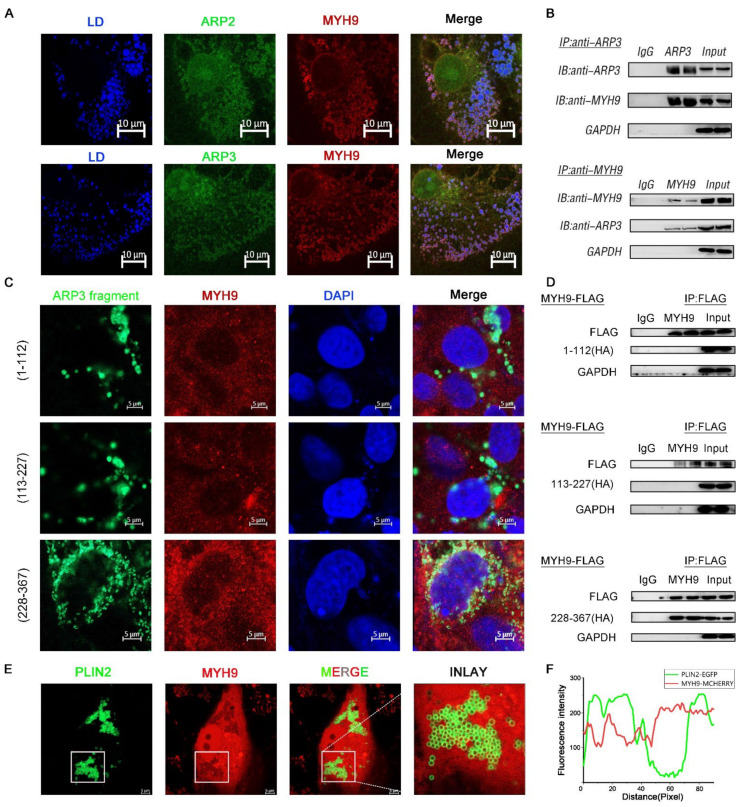
MYH9 interacted with ARP3 for LD movement. (**A**) Subcellular localization of ARP2 or ARP3 and MYH9, bar, 10 μm (**B**) The binding of ARP3 and MYH9 detected by co-immunoprecipitation. (**C**) Subcellular localization of ARP3 truncation fragments and MYH9, bar, 5 μm. (**D**) The binding of ARP3 truncation fragments and MYH9 detected by co-immunoprecipitation. (**E**) Subcellular localization of Plin2 and MYH9, bar, 5 μm. (**F**) The Fluorescence intensity analysis of (**E**).

**Figure 4 ijms-23-08730-f004:**
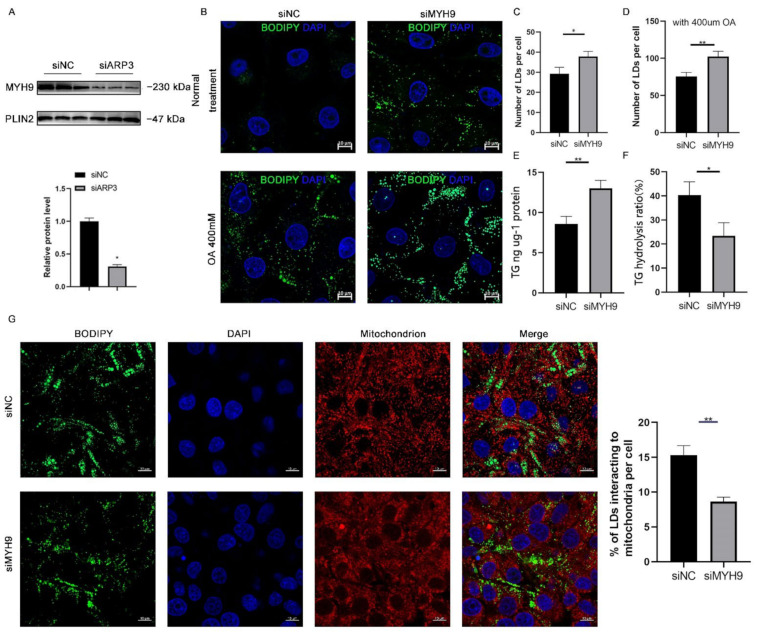
MYH9 regulates LD content. (**A**) The expression of MYH9 of isolated cLDs after interfering with ARP3 and control cells was detected by Western Blot experiments. (**B**) Detection of the cellular LDs marked by BODIPY493/503 through a laser scanning confocal microscope. The Hela cells were transfected with siNC or siMYH9 for 48 h. Subsequently, the cells were treated with 400 μM oleic acid for another 6 h. Then, the cells were fixed and stained for observation by microscope. bar, 10 μm. (**C**,**D**) The gray value analysis of the difference in the number of LDs. * *p* < 0.05, ** *p* < 0.01. (**E**,**F**) Analysis of the difference in the content of triglycerides and hydrolysis rate after interference MYH9. * *p* < 0.05, ** *p* < 0.01. (**G**) Detection of the interaction of LDs and mitochondrion after siMYH9. ** *p* < 0.01.

**Figure 5 ijms-23-08730-f005:**
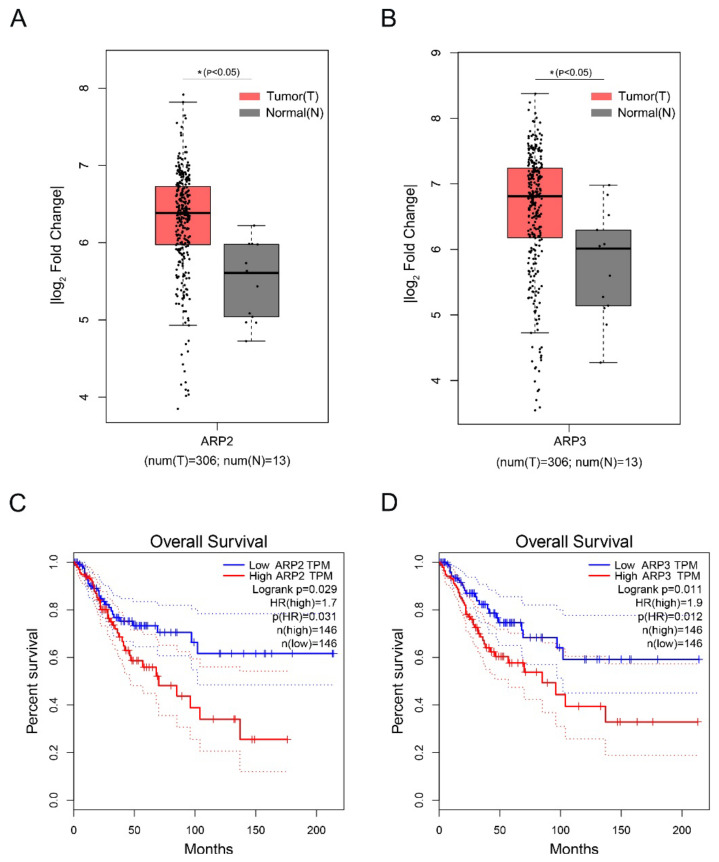
The high expression of ARP2 and ARP3 is associated with a poor prognosis of Cervical Squamous Cell Carcinoma. (**A**,**B**) Box plot of expression of ARP2 and ARP3 in CSCC. Gene expression analysis and survival analysis were performed by the GEPIA database. (**C**,**D**) The survival analysis of ARP2 and ARP3 in CSCC. Gene expression analysis and survival analysis were performed by the GEPIA database.

## Data Availability

Not applicable.

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
