# Peer review of "ARP2/3 Regulates Fatty Acid Synthesis by Modulating Lipid Droplets’ Motility"

_ijms, 2022, doi:10.3390/ijms23158730_

Round 1
Reviewer 1 Report
It is well established that LD physically and functionally interacts with cellular organelles, including the ER, mitochondria, lysosomes, and peroxisomes. The LD -organelle contacts are in part controlled by LD intracellular motility. LDs can be transported directly by motor proteins along either actin filaments or microtubules. In this study, Han and co-workers in my knowledge for the first time examined the function of microfilament-associated proteins 2 and 3 (ARP2 and ARP3) in regulating LDs motility using Hela cells as a model. Well understanding the interaction between components of the cytoskeleton in LD indeed could provide insight into carcinogenesis.
The introduction was well written. However, in the last paragraph (line55-57) references are missing
The changes in TG content, TG hydrolysis ratio, and a number of contact sites with mitochondria are in accordance with other studies showing that upon SREBF1 inhibition decreased triglyceride accumulation was observed paralleled by an increase in mitochondrial respiration (Audet-Walsh et al 2018). Measurement of respiration would further support the findings. Data concerning the content of sterols and sterol esters would be also of interest. Fig 5 A and B missing the statistical analysis.
The discussion needs to be rewritten to focus on the new finding presented in the current report. Also referencing the figures showing the discussed data would help to follow.
Material and methods missing the paragraph describing the measurement of TG and TG hydrolysis rate.
Author Response
It is well established that LD physically and functionally interacts with cellular organelles, including the ER, mitochondria, lysosomes, and peroxisomes. The LD -organelle contacts are in part controlled by LD intracellular motility. LDs can be transported directly by motor proteins along either actin filaments or microtubules. In this study, Han and co-workers in my knowledge for the first time examined the function of microfilament-associated proteins 2 and 3 (ARP2 and ARP3) in regulating LDs motility using Hela cells as a model. Well understanding the interaction between components of the cytoskeleton in LD indeed could provide insight into carcinogenesis.
The introduction was well written. However, in the last paragraph (line55-57) references are missing
Response: According to the reviewer’s suggestion, we have added a related literature in the reference part (L55-L59, Ref.16-19
- Bersuker, K.; Olzmann, J.A. Establishing the lipid droplet proteome: Mechanisms of lipid droplet protein targeting and deg-radation. Biochim Biophys Acta Mol Cell Biol Lipids 2017, 1862, 1166-1177, doi:10.1016/j.bbalip.2017.06.006.
- Roberts, M.A.; Olzmann, J.A. Protein Quality Control and Lipid Droplet Metabolism. Annu Rev Cell Dev Bi 2020, 36, 115-139, doi:10.1146/annurev-cellbio-031320-101827.
- Hamp, J.; Lower, A.; Dottermusch-Heidel, C.; Beck, L.; Moussian, B.; Flotenmeyer, M.; Onel, S.F. Drosophila Kette coor-dinates myoblast junction dissolution and the ratio of Scar-to-WASp during myoblast fusion. J Cell Sci 2016, 129, 3426-3436, doi:10.1242/jcs.175638.
- Romani, P.; Nirchio, N.; Arboit, M.; Barbieri, V.; Tosi, A.; Michielin, F.; Shibuya, S.; Benoist, T.; Wu, D.; Hindmarch, C., et al. Mitochondrial fission links ECM mechanotransduction to metabolic redox homeostasis and metastatic chemotherapy resistance. 2022.)
The changes in TG content, TG hydrolysis ratio, and a number of contact sites with mitochondria are in accordance with other studies showing that upon SREBF1 inhibition decreased triglyceride accumulation was observed paralleled by an increase in mitochondrial respiration (Audet-Walsh et al 2018). Measurement of respiration would further support the findings. Data concerning the content of sterols and sterol esters would be also of interest. Fig 5 A and B missing the statistical analysis.
Response: According to the reviewer’s suggestion, we have added a related literature in the reference part (L259, Ref.31 Audet-Walsh, É.; Vernier, M.; Yee, T.; Laflamme, C.; Li, S.; Chen, Y.; Giguère, V. SREBF1 Activity Is Regulated by an AR/mTOR Nuclear Axis in Prostate Cancer. Molecular cancer research : MCR 2018, 16.) and performed the statistical analysis of Fig 5
The discussion needs to be rewritten to focus on the new finding presented in the current report. Also referencing the figures showing the discussed data would help to follow.
Response: According to the reviewer’s suggestion, we re-write the discussion part.L219-269
Material and methods missing the paragraph describing the measurement of TG and TG hydrolysis rate.
Response: According to the reviewer’s suggestion, the methods about the measurement of TG and TG hydrolysis rate is now provided in L365-L374.
Reviewer 2 Report
Major Issues to Address
1. The manuscript lacks sufficient explanation of the methods. In particular, the authors should
a) Clarify which experiments included oleic acid treatment
b) Include methods for using LipidTox in section 4.4
c) Include methods for LD fractionation
d) Indicate which antibodies were used (section 4.5)
e) Explain the method for measuring TG hydrolysis (for Figure 4F)
f) Clarify the methods for Fluorescent Image Analysis (section 4.6). The manuscript refers to a dotted line in Figure 3C, 3D which isn't understandable to this reviewer. Also, authors should explain their method for measuring LD/mitochondria interactions.
2. The proposed model isn’t adequately supported by the data. Specific issues include:
a) Abstract describes “number and size of LDs was inhibited after ARP2/3 interfered” (Line 16). The effect on LDs is not shown in siARP3 treated cells, but instead following siMYH9 treatment (Fig 4BCD). The authors are likely assuming that knock-down of MYH9 leads to ARP2/3 interference, but this remains conjecture. Moreover, only LD number is quantified following siMYH9 treatment. LD size is not discussed. .
b) Abstract states that “MYH9 regulated LDs content by binding with LDs and ARP2/3” (Line 15-16). This conclusion overstates the results that are shown. The only suggestion of possible change in LD content is the increase in TG with siMYH9 (Figure 4E), a result which is likely due to the increase in LD number and not content of any individual LD.
c) Figure 1A does not show colocalization between ARP3, actin, and LDs. The authors should consider an inset or zoomed portion of the figure to better highlight this result.
d) Figure 3E does not support MYH9 on LDs. Instead, MyH9 staining occurs throughout the cell.
e) Figure 4A suggests less MYH9 in LD fractions from siARP3 treated cells. Because the authors haven’t explained their method for LD fractionation, nor demonstrated convincingly that MYH9 specifically interacts with LDs (in Fig 3E), it’s possible that this decrease is indicative of an overall decrease in MYH9. How does siARP3 change MYH9 expression?
f) Figure 4G. Mitochondrial staining in the siMYH9 cells looks fainter. Could this have influenced the quantification? Is there a difference in mitochondrial number with siMYH9?
g) The model presented in section 5. Conclusions again overstates the results. For example, there is no evidence that knock-down of ARP2/3 diminished LD motility (line 347), nor that “overexpression led to the opposite result” (line 348). (Only gene expression was examined after overexpression.)
Additional Questions and Suggesstions
1. This reviewer is confused by the ARP3 fragments from Figure 1FGH and Figure 3CD. Is the 1-112aa the same as the ARP3-deltaN? And if so is this ARP3 missing the N terminus (as suggested by the ARP3-deltaN) or is it the N-terminal fragment only (as suggested by the 1-112aa notation)? If these fragments are only the 1-112 or 113-227 or 228-367 domains, then the deltaN or M or C notation in Figure 1F is confusing. Stick with one notation for these domains.
2.This reviewer is confused by the apparent difference in localization of the ARP3 fragment 1-112 in Figure 3C and 1G and 1H. What is different about these experiments?
3. Data in Figure 2G-J is not discussed in the results of this manuscript, nor is it explained why these results are relevant to this paper.
4. Line 136-137: “The movement of LDs depended on the cytoskeleton and required motor proteins to provide power.” As this paper hasn’t directly looked at LD movement, I assume this statement is referring to previously published work. In which case this statement needs a reference.
5. Line 167: Should refer to Figure 4A
Author Response
Response to Reviewer 2 Comments
Reviewer 2
- The manuscript lacks sufficient explanation of the methods. In particular, the authors should
- a) Clarify which experiments included oleic acid treatment
- b) Include methods for using LipidTox in section 4.4
- c) Include methods for LD fractionation
- d) Indicate which antibodies were used (section 4.5)
- e) Explain the method for measuring TG hydrolysis (for Figure 4F)
- f) Clarify the methods for Fluorescent Image Analysis (section 4.6). The manuscript refers to a dotted line in Figure 3C, 3D which isn't understandable to this reviewer. Also, authors should explain their method for measuring LD/mitochondria interactions.
Response:
- a) Only in section 2.4, oleic acid treatment was performed when interfere MYH9 (Fig 4B and D), the unlabeled experiments did not include oleic acid treatment. We have specially labeled in the figure and L283-284
- b) According to the reviewer’s suggestion, we have detailed explanation in L309-313
- c) According to the reviewer’s suggestion, we have detailed explanation in L376-381 for LDs isolation.
- d) According to the reviewer’s suggestion, we have detailed explanation in L342-348 for antibodies
- e) According to the reviewer’s suggestion, we have detailed explanation in L365-L374 the measurement of TG and TG hydrolysis rate.
- f) According to the reviewer’s suggestion, we redrew the picture and have detailed explanation in L318-321.
- The proposed model isn’t adequately supported by the data. Specific issues include:
- a) Abstract describes “number and size of LDs was inhibited after ARP2/3 interfered” (Line 16). The effect on LDs is not shown in siARP3 treated cells, but instead following siMYH9 treatment (Fig 4BCD). The authors are likely assuming that knock-down of MYH9 leads to ARP2/3 interference, but this remains conjecture. Moreover, only LD number is quantified following siMYH9 treatment. LD size is not discussed. .
Response: Sorry, what we want to express is that the number of lipid droplets and cellular lipid content are increased after MYH9 interference. According to the reviewer’s suggestion, we rewrite this sentence. L16
- b) Abstract states that “MYH9 regulated LDs content by binding with LDs and ARP2/3” (Line 15-16). This conclusion overstates the results that are shown. The only suggestion of possible change in LD content is the increase in TG with siMYH9 (Figure 4E), a result which is likely due to the increase in LD number and not content of any individual LD.
Response: Done as requested by the reviewer. L16
- c) Figure 1A does not show colocalization between ARP3, actin, and LDs. The authors should consider an inset or zoomed portion of the figure to better highlight this result.
Response: Done as requested by the reviewer. We added a zoomed-in image and marked it with white arrows.
- d) Figure 3E does not support MYH9 on LDs. Instead, MyH9 staining occurs throughout the cell.
Response: The diffuse distribution of MYH9 is consistent with what has been reported in the literature(Silencing MYH9 blocks HBx-induced GSK3β ubiquitination and degradation to inhibit tumor stemness in hepatocellular carcinoma). We suggest that MYH9 is not only involved in the metabolic activity of LD, and we detected MYH9 on intracellular lipid droplets. These results demonstrate that MYH9 interacts with LDs.
- e) Figure 4A suggests less MYH9 in LD fractions from siARP3 treated cells. Because the authors haven’t explained their method for LD fractionation, nor demonstrated convincingly that MYH9 specifically interacts with LDs (in Fig 3E), it’s possible that this decrease is indicative of an overall decrease in MYH9. How does siARP3 change MYH9 expression?
Response: According to the reviewer’s suggestion, we supplemented the method of LDs extraction L376-381. MYH9 is diffusely distributed in cells, and we believe that such a protein is not solely involved in LDS metabolism. By fluorescence localization and western blot of isolated LDs, we demonstrated that MYH9 can interact with LD (Fig3A, 4A). The mechanism how siARP3 affects MYH9 expression is not clear. They are important issues that we will tackle in our next piece of work.
- f) Figure 4G. Mitochondrial staining in the siMYH9 cells looks fainter. Could this have influenced the quantification? Is there a difference in mitochondrial number with siMYH9?
Response: This does not affect the specific quantification. We examined the number of mitochondria after SiMYH9 and the results did not change significantly (data not shown)。
- g) The model presented in section 5. Conclusions again overstates the results. For example, there is no evidence that knock-down of ARP2/3 diminished LD motility (line 347), nor that “overexpression led to the opposite result” (line 348). (Only gene expression was examined after overexpression.)
Response: We thank the reviewer for their helpful suggestions and rewrite this sentence. L396-L397
Additional Questions and Suggesstions
- This reviewer is confused by the ARP3 fragments from Figure 1FGH and Figure 3CD. Is the 1-112aa the same as the ARP3-deltaN? And if so is this ARP3 missing the N terminus (as suggested by the ARP3-deltaN) or is it the N-terminal fragment only (as suggested by the 1-112aa notation)? If these fragments are only the 1-112 or 113-227 or 228-367 domains, then the deltaN or M or C notation in Figure 1F is confusing. Stick with one notation for these domains.
Response: According to the reviewer’s suggestion. The full text has been unified as 1-112aa, and the delta symbol has been removed.
2.This reviewer is confused by the apparent difference in localization of the ARP3 fragment 1-112 in Figure 3C and 1G and 1H. What is different about these experiments?
Response: According to the reviewer's suggestion, we reorganized the results and the arp3 fragment 1-56aa region has a more precise microfilament localization.
- Data in Figure 2G-J is not discussed in the results of this manuscript, nor is it explained why these results are relevant to this paper.
Response: Done as requested by the reviewer. L123-L129
- Line 136-137: “The movement of LDs depended on the cytoskeleton and required motor proteins to provide power.” As this paper hasn’t directly looked at LD movement, I assume this statement is referring to previously published work. In which case this statement needs a reference.
Response: According to the reviewer’s suggestion, we have added a related literature in the reference part (Ref. 22 George, O.; Hongyun, W. Rotary protein motors. Trends Cell Biol 2003, 13.)
- Line 167: Should refer to Figure 4A
Response: Done as requested by the reviewer.

Reviewer 3 Report
This paper by Han and coworkers argues that ARP2/3 links lipid droplets to myosins and regulates mobility of the lipid droplets with consequent effects on fatty acid synthesis. Data includes localization of the various components being analyzed as well as truncation and siRNA experiments. I found the overall analysis logical and interesting. However, there are specific points where additional clarification is needed.
1. In Figure 1, association of LDs with actin filaments is analyzed quantitatively in panels (D) and ( E). Localization of LDs in cells expressing various Arp3 fragments and of the Arp3 truncations themselves is shown in (G) and (H). The N-terminal truncation clearly associates with actin filaments in (G) and with LDs in (H) but the data in this figure by itself is insufficient to conclude that the statement in lines 88 and 89 of page two “The results showed that the N-terminal region . . . was the key domain mediated the interaction between LDs and microfilaments” is correct. It is better reworded to “These results are consistent with the N-terminal region of ARP3(1-112aa) being the key domain mediating the interaction between LDs and microfilaments.”
2. In Figure 1F, the name ARP3-deltaN implies to me that the N-terminal region is deleted while the other regions remain when it is the reverse that is actually true. I think the naming scheme in (G) and (H) (1-112aa) is better and should be used in (F) also. Same for the other two fragments.
3. In Figure 2G, the 113-227aa fragment localizes to structures that might be mitochondria. Could this be checked by staining for a mitochondrial marker like Tom20, and are there reasons to suggest this could be the case given there are previous studies that suggest ARP2/3 can associate with mitochondria?
4. In Figure 3A there is convincing colocalization of ARP2 and MYH9 to the rims of LDs, which would be expected for truly LD-associated proteins. I think this should be mentioned in section 2.3.
5. In 4G, % of LDs interacting with mitochondria is given. There should be an explanation in Methods as to how this is quantitated.
6. There are a number of small errors in the English throughout the paper. These generally don’t affect understanding and could be fixed without much difficulty by an editing service employing native English speakers.
7. Given that mitochondria are brought up in the Discussion, are known to interact with LDs and there is some previous literature related to ARP2/3 on mitochondria, I think there should be a few sentences touching on this added to the Introduction.
Author Response
Reviewer 3
This paper by Han and coworkers argues that ARP2/3 links lipid droplets to myosins and regulates mobility of the lipid droplets with consequent effects on fatty acid synthesis. Data includes localization of the various components being analyzed as well as truncation and siRNA experiments. I found the overall analysis logical and interesting. However, there are specific points where additional clarification is needed.
- In Figure 1, association of LDs with actin filaments is analyzed quantitatively in panels (D) and ( E). Localization of LDs in cells expressing various Arp3 fragments and of the Arp3 truncations themselves is shown in (G) and (H). The N-terminal truncation clearly associates with actin filaments in (G) and with LDs in (H) but the data in this figure by itself is insufficient to conclude that the statement in lines 88 and 89 of page two “The results showed that the N-terminal region . . . was the key domain mediated the interaction between LDs and microfilaments” is correct. It is better reworded to “These results are consistent with the N-terminal region of ARP3(1-112aa) being the key domain mediating the interaction between LDs and microfilaments.”
Response: We thank the reviewer for their helpful suggestions and rewrite this sentence according to reviewer’s suggestion.
- In Figure 1F, the name ARP3-deltaN implies to me that the N-terminal region is deleted while the other regions remain when it is the reverse that is actually true. I think the naming scheme in (G) and (H) (1-112aa) is better and should be used in (F) also. Same for the other two fragments.
Response: We thank the reviewer for their helpful suggestions and rewrite this sentence according to reviewer’s suggestion.
- In Figure 2G, the 113-227aa fragment localizes to structures that might be mitochondria. Could this be checked by staining for a mitochondrial marker like Tom20, and are there reasons to suggest this could be the case given there are previous studies that suggest ARP2/3 can associate with mitochondria?
Response: We thank the reviewers for their suggestions, but we are very sorry that we did not detect the co-localization of ARP2/3 with mitochondria. The reviewer raised a very interesting idea and the mechanism how ARP2/3 affects LDs movement is the next goal in our research.
- In Figure 3A there is convincing colocalization of ARP2 and MYH9 to the rims of LDs, which would be expected for truly LD-associated proteins. I think this should be mentioned in section 2.3.
Response: We thank the reviewers for their suggestions and we have revised them accordingly. L146-152
- In 4G, % of LDs interacting with mitochondria is given. There should be an explanation in Methods as to how this is quantitated.
Response: According to the reviewer’s suggestion, we redrew the picture and have detailed explanation in L318-321.
- There are a number of small errors in the English throughout the paper. These generally don’t affect understanding and could be fixed without much difficulty by an editing service employing native English speakers.
Response: We thank the reviewers for their suggestions and we have revised them accordingly.
- Given that mitochondria are brought up in the Discussion, are known to interact with LDs and there is some previous literature related to ARP2/3 on mitochondria, I think there should be a few sentences touching on this added to the Introduction.
Response: We thank the reviewers for their suggestions and we have revised them accordingly. L55-59
Round 2
Reviewer 1 Report
The authors addressed the reviewer's concerns and improved the manuscript.